# Production and Characterization of Cellulose Nanocrystals from Eucalyptus Dissolving Pulp Using Endoglucanases from *Myceliophthora thermophila*

**DOI:** 10.3390/ijms241310676

**Published:** 2023-06-26

**Authors:** Pratima Waghmare, Nuo Xu, Pankajkumar Waghmare, Guodong Liu, Yinbo Qu, Xuezhi Li, Jian Zhao

**Affiliations:** State Key Laboratory of Microbial Technology, Shandong University, Qingdao 266237, China; pratimapw22@gmail.com (P.W.); xn_17735576963@163.com (N.X.); wpankaj97@sdu.edu.cn (P.W.); gdliu@sdu.edu.cn (G.L.); quyinbo@sdu.edu.cn (Y.Q.); lixz@sdu.edu.cn (X.L.)

**Keywords:** endoglucanases, enzymatic characteristics, cellulose nanocrystals, enzymatic hydrolysis, *Myceliophthora thermophila*, eucalyptus dissolving pulp

## Abstract

Endoglucanase (EG) is a key enzyme during enzymatic preparation of cellulose nanocrystals (CNCs). *Myceliophthora thermophila* is a thermophilic fungus that has thermal properties and a high secretion of endoglucanases (EGs), and could serve as potential sources of EGs for the preparation of CNCs. In this work, four different GH families (GH5, GH7, GH12, and GH45) of EGs from *M. thermophila* were expressed and purified, and their enzymatic characteristics and feasibility of application in CNC preparation were investigated. It was shown that the MtEG5A from *M. thermophila* has good potential in the enzymatic preparation of CNCs using eucalyptus dissolving pulp as feedstock. It was also observed that there was a synergistic effect between the MtEG5A and other MtEGs in the preparation of CNCs, which improved the yield and properties of CNCs obtained by enzymatic hydrolysis. This study provides a reference for understanding the enzymatic characteristics of different families of EGs from *M. thermophile* and their potential application in nanocellulose production.

## 1. Introduction

In the current global scenario, the research and development of renewable, sustainable, and environmentally friendly materials have gained much attention. Nanocellulose derived from lignocellulose is a bio-based nanomaterial with at least one nanoscale dimension (1–100 nm) [1] and has several advantageous properties, such as a unique morphology, nanoscale dimension, and excellent mechanical properties [2]. Thus, it has great potential for application in various industries, such as papermaking [3], packaging [4], coating additives [5], medical applications, preparation of antibacterial agents, skincare, etc. [6,7]. Based on preparation techniques, nanocellulose can be classified as cellulose nanocrystals (CNCs), cellulose nanofibers (CNFs), and bacterial nanocellulose (BNC) [8].

CNCs are also referred to as nanocrystalline cellulose, cellulose nano-whiskers, and crystalline nanocellulose. Depending on the sources of raw materials and preparation methods, CNCs can have a whisker shape, a short rod-like shape, a spindle shape, or a spherical shape. The size of CNCs highly depends on the cellulose sources, usually with a length in the range of 100 nm to several micrometers [9] and a width of 2–20 nm with a high degree of crystallinity [7]. CNCs isolated from tunicates and bacteria cellulose have a high aspect ratio (>50), whereas CNCs obtained from cotton and wood pulp have aspect ratios below 20 [10]. Moreover, CNCs have high thermal stability (about 300 °C) and shear-thinning behavior in aqueous suspensions [8,11]. CNCs are usually prepared by acid hydrolysis [2,12], TEMPO-mediated oxidation [13], alkali methods [14], mechanical disintegration, and enzymatic treatment methods [15,16,17]. Among them, acid hydrolysis is the most common method for CNC preparation, but involves serious wastewater problems and high device cost because of highly corrosive acid solutions. The enzymatic hydrolysis method is gaining more interest due to several operational advantages, such as no requirement for corrosion-resistant equipment, being environmentally friendly, using milder operating conditions, and formation of less undesirable by-products. Furthermore, the greater specificity of biocatalysts results in the production of CNCs with different morphologies [7].

The enzymes used in the enzymatic hydrolysis of lignocellulose include carbohydrate-active enzymes, especially cellulases, a multienzyme complex including endoglucanases (EC 3.2.1.4), exoglucanases, or cellobiohydrolases (EC 3.2.1.91), and β-glucosidase (EC 3.2.1.21), which act synergistically to complete the degradation of lignocellulose [18]. During the enzymatic preparation of CNCs, EGs are a major enzyme and it randomly act on the amorphous region of cellulose [10]. From the literature data, however, there is a lack of commercial enzymatic preparations that are specifically designed for the production of nanocellulose so far. CNCs produced by controlled enzymatic hydrolysis made use of commercial cellulase preparations such as Celluclast 1.5 L, Cellusoft L, and other cellulase cocktails that were designed for complete hydrolysis of cellulose into fermentable sugars [19,20]. Commercial cellulase preparations are not the most suitable for CNC preparation [21]. A possible disadvantage of using the commercial cellulase preparations for the isolation of CNCs is that it may fail to make use of potential endoglucanases’ selectivity for the amorphous regions of cellulose, due to the presence of cellobiohydrolases (crystalline depolymerizing enzymes) and β-glucosidase in significant amounts in such preparations.

To date, many articles have reported the use of endoglucanases for the production of nanocellulose; for example, Siqueira et al. [21] investigated the action of endoglucanases from fungal sources such as EG GH7-TL from *Trichoderma longibrachiatum* (Megazyme), EG GH45-FC (FiberCare R, Novozymes), and bacterial sources such as EG GH5-BA from *Bacillus amyloliquefaciens*, using bleached eucalyptus kraft pulp as the substrate. Wang et al. [22] studied pre-treatment with hyper thermostable endoglucanase (PhGH5-archaea) combined with micro fluidization to produce CNFs, which resulted in a 30% reduction in mechanical energy input compared with fibers without enzymatic treatment. Nechyporchuk et al. [23] produced CNFs from bisulfite softwood pulp through enzymatic pre-treatment and TEMPO-oxidation followed by wet grinding; in this enzymatic pretreatment, different enzyme formulations, such as commercial enzyme solution and mono-component endoglucanase, were studied and compared. Xu et al. [24] used endoglucanase derived from *Aspergillus oryzae* to prepare CNCs by an enzyme-assisted process from natural bast fibers; the size distribution of CNCs from flax fibers appeared narrow due to a specific enzyme digestion site. Although the effects of some EGs on nanocellulose production have been evaluated, the EGs from different sources and families have different characteristics, which affect the efficiency of the preparation process. In our lab’s previous work, four EGs from *Penicillium oxalicum* were expressed, and their effects on CNC preparation by hydrolyzing MCC were investigated. It was found that the mixture of rPoCel7B and rPoCel5B achieved the highest yield of CNCs with a more dispersed and uniform morphology [17]. Therefore, it is still necessary to assess the effects of various sources of EGs on CNC preparation for screening more suitable EGs to achieve enzymatic production of CNCs.

*Myceliophthora thermophila* is a thermophilic filamentous fungus, which is classified as an ascomycete, that contains an extremely powerful cellulolytic microorganism that synthesizes a complete set of enzymes necessary for the degradation of cellulose [25]. The genome of this fungus comprises eight nucleotide sequences with EG activity, distributed among different glycoside hydrolase (GH) families (5, 7, 12, 45), all of which are predicted to be extracellular and possess a catalytic domain and several *N*- and *O*-glycosylation sites [26]. The fungal EGs belonging to families GH5 and GH7 cleave the glycosidic bond of their substrates using the double-displacement retaining mechanism and have highest saccharification activity, for example, MtEG5A and MtEG7A, which are 42 and 48 kDa enzymes, respectively [26,27]. The molecular weights of MtEG12A and MtGH45A, which belong to families GH12 and GH45, respectively, are relatively low and they do not possess a CBM region. MtEG45A shares similarities with endoglucanases derived from termites and archaea, which break similar linkages to those cleaved by cellobiohydrolases, suggesting that this enzyme is substrate-dependent and can perform endo- and exo-mechanisms of action [28]. In this context, the secretions of abundant types of EG enzymes from *M. thermophila* are expected to be a potential source of EGs for nanocellulose preparation. On the other hand, it was also reported the EGs from *M. thermophila* were heat resistant; for example, MtEG5A, MtEG7A, and MtEG45A showed optimum temperatures of 70 °C, 60 °C, and 65 °C, respectively. After pre-incubation in 100 mM phosphate–citrate buffer (pH 5.0) for 8 h, MtEG5A at 50 °C and MtEG7A at 40 °C remained fairly stable, whereas MtEG45A was stable up to 24 h at temperatures up to 70 °C [26,27,28,29]; therefore, these enzymes are expected to accelerate the preparation of CNCs by enzymatic hydrolysis at a relatively high temperature due to their temperature resistance and stability characteristics.

In this study, several different families of EGs from *M. thermophila* were expressed in the P. oxalicum M12 strain and purified, and their enzymatic characteristics were investigated. Based on them, the feasibility of applying the EGs in nanocellulose preparations was assessed by using eucalyptus dissolving pulp (EDP) as a cellulosic feedstock due to its high-purity cellulose content. After screening potential endoglucanases, we attempted to enhance the properties of the CNCs using a combination of the endoglucanases. The obtained CNCs were characterized by transmission electron microscopy (TEM), scanning electron microscopy (SEM), X-ray diffraction (XRD), Fourier Transform Infrared Spectroscopy (FT-IR), and thermogravimetric analysis (TG). This study provides a reference for understanding the enzymatic characteristics of different families of EGs from *M. thermophila* and their potential application in nanocellulose production.

## 2. Results and Discussion

### 2.1. Chemical Composition of EDP

The chemical composition of the EDP that was used as the cellulosic substrate for producing CNCs was first determined and is shown in Table 1. It was found that there was very high percentage content of cellulose in the EDP (91.45%), and a very low amount of hemicellulose (2.23%) and a trace amount of lignin (0.72%), indicating that the EDP was a high-purity cellulose substrate and suitable for studying the effect of different EGs on CNC preparation. In previous studies, microcrystalline cellulose (MCC), a cellulosic feedstock with 99% cellulose content, was usually used as the substrate to investigate the enzymatic preparation of CNCs with cellulase [17,30], but its cost is higher than that of EDP, which affects the economics of CNC preparation. In addition, the crystallinity of MCC is higher than that of EDP, which influences the enzymatic hydrolysis efficiency of EGs acting on the amorphous region of cellulose.

### 2.2. Expression and Purification of Different EGs Derived from M. thermophila

Genomic DNA of *M. thermophila* was extracted and used to amplified the cDNA of different GH family endoglucanases using a gene-specific primers, which were further cloned into the constitutive vector P*ubiD* (Appendix A) for heterologous expression in *P. oxalicum* (M12 host). Table 2 shows the general information of the endoglucanases from *M. thermophila*. The correct transformants were confirmed by PCR and sequencing. The constitutive promoter produces relatively pure protein and provides a high expression level of the target gene [31,32]. All the secreted proteins were further purified using HisTrap™ FF columns and SDS-PAGE analysis was used to determine the purity of the proteins (Appendix A).

The specific activity analysis showed that the maximum specific activity was found in MtEG5A and the lowest was in MtEG12A (Table 2); this could be due to the structural difference between the enzymes’ active sites [33]. Usually, the molecular weights of endoglucanases from fungi are in the range of 25–75 kDa [34]. It was found by SDS-PAGE (Appendix A) that the molecular weights of the pure MtEG5A, MtEG7A, MtEG12A, and MtEG45A proteins were 50, 49, 30, and 28 kDa, respectively, which are within the molecular weight range of fungal endoglucanases reported in the literature, but larger than the theoretical value predicated using the ExPASy-Compute pI/Mw tool (https://web.expasy.org/compute_pi/) (accessed on 10 March 2023) (Table 2); this could be due to potential post-translation modifications, including *N*- and *O*-glycosylation at the Asn-Xaa-Ser/Thr sequences of the expressed protein since there are 17, 16, and 3 predicted *O*-glycosylation sites for MtEG5A, MtEG7A, and MtEG45A, respectively, as shown in Table 2.

### 2.3. Enzymatic Properties of Different EGs from M. thermophila

The purified proteins were used to investigate the enzymatic properties of the different EGs from *M. thermophila*. Figure 1a shows the optimum temperatures of MtEG5A, MtEG7A, MtEG12A, and MtEG45A were 75 °C, 70 °C, 65 °C, and 65 °C, respectively. The optimum temperatures of MtEG5A and the MtEG7A showed slight variations compared to the previously reported optimum temperatures of EG 5 and EG 7 (70 °C and 60 °C, respectively) [26,27,29], which may be due to the difference in the expression host.

Figure 1b shows that the optimum pH for MtEG5A, MtEG7A, and MtEG12A was pH 5.0, and MtEG45A showed the highest activity levels at pH 4.0. For MtEG5A, MtEG7A, and MtEG45A, there was >70% of the peak activity at pH 6.0 and >60% of activity was shown at pH 7.0; for MtEG12A, the enzymatic activity dropped rapidly when the pH was less than 5.0 or higher than 7.0.

The thermostability of all the purified recombinant endoglucanases was determined at 50 °C, 55 °C, 60 °C, and 65 °C, and results are shown in Appendix A. It was found that MtEG5A and MtEG7A remained highly stable below 65 °C, and MtEG7A had a relatively higher thermal stability than MtEG5A. More than 90% of the activity remained after pre-incubation for 3 h below 60 °C for MtEG5A and MtEG7A (Appendix A); at 65 °C, MtEG5A and MtEG7A still retained about 60% and about 80% enzymatic activity, respectively (Appendix A). The thermal stability of MtEG5A and MtEG7A was comparable to the previously reported thermostable endoglucanase [26,27]. In contrast, MtEG12A and MtEG45A had low thermal stability compared to MtEG5A and MtEG7A, especially MtEG12A. After an incubation of 1 h at 65 °C, just about 20% activity remained for MtEG45A, and for MtEG12A, the relative activity was less than 10% (Appendix A).

### 2.4. Effect of Different EGs and EG Combinations on the Preparation of CNCs

EGs from different sources and families show different enzymatic characteristics which affects the process efficiency of CNC preparation [17]. Here, the feasibility of different endoglucanases from *M. thermophila* in CNCs production was preliminarily assessed by using the EGs to hydrolyze EDP. The sample obtained by the same enzymatic process under identical conditions but without enzymes was used as a control. Figure 2a shows the size distribution pattern of the CNCs prepared by enzymatic hydrolysis with the different EGs. It was found that, compared to MtEG5A, the other three EGs, including MtEG7A, MtEG12A, and MtEG45A, produced CNCs with relatively big particle sizes and a wide size distribution, but the CNCs obtained by enzymatic hydrolysis of EDP with MtEG5A showed only a single and sharp peak in the range below 800 nm, indicated that the CNCs had a relatively uniform size.

Figure 2b shows that there the lowest yield of CNCs by enzymatic hydrolysis was with MtEG12A, which may be partly due to the substrate specificity of EG12 (Vlasenko, 2010). It was reported that EG12 displays high activity against β-glucan, xyloglucan, and lichenan, indicated it is a nonspecific glycosyl hydrolase and has diverse specific activity against a set of polysaccharides [33,35]. Our experimental results also proved that MtEG12A was more specific toward barley β-glucan, xyloglucan, and lichenan, but was less specific toward CMC. Figure 2a also shows that there were many particles with a size exceeding 1 μm in the suspension of CNCs obtained by enzymatic hydrolysis with MtEG12A, indicating its low ability to hydrolyze cellulose from one perspective. The higher yield of CNCs obtained by MtEG7A and MtEG45A than MtEG5A may be attributed to the relatively larger particles present in the CNC suspension because the CNC yield was determined by gravimetric analysis.

As the CNCs obtained by enzymatic hydrolysis with MtEG5A had a more uniform and relatively small particle size, different enzyme combinations including MtEG5A, named (5+7), (5+12), and (5+7+45), were used to hydrolyze EDP to prepare CNCs. Here, 5, 7, 12, and 45 represent the endoglucanases from different GH families. Appendix A shows SEM images of the obtained CNCs, and their length and width revealed using ImageJ software (version 1.8.0) are shown in Table 3. The average apparent size of the CNC samples determined by DLS analyses is also listed in Table 3.

It was shown that, compared to MtEG5A alone, different EG combinations produced CNCs with smaller sizes, showing a synergistic effect between EG5 and other different GH family endoglucanases; in particular, the EG combination of (5+7) produced the CNCs with the smallest particle size, suggesting that the synergistic effect of MtEG5 and MtEG7 was relatively more significant. All the results were consistent with the conclusion obtained from previous work performed in our lab [17]. Both EG5 and EG7 are retaining enzymes, and they have a CBM region and higher specific activity compared to EG12 and EG45, which may be part of the reason why EG5 and EG7 showed a strong synergistic effect during enzymatic hydrolysis of a cellulosic substrate. On the other hand, although it was known that Dynamic Light Scattering (DLS) does not accurately measure CNC particle size, the average apparent sizes obtained from the DLS analyses (Table 3) confirmed that DLS, as a fast and effective method, can still be used for the determining CNC particle size for preliminary assessments of the effects of EGs on the enzymatic preparation of CNCs.

Siqueira et al. [21] prepared nanocellulose with length of 0.2–1.5 um and width of 4 to 30 nm from eucalyptus pulp by an enzymatic method. Zhu et al. [36] obtained CNCs with a length of 500 nm and diameter of 20 nm by enzymatic hydrolysis combined with a microfluidizer. Cui et al. [37] produced CNCs with a length of 500–750 nm and width of 5–8 nm using ultrasound-assisted enzymatic hydrolysis. Filson et al. [38] used commercial cellulose preparations to hydrolyze recycled pulp and obtained nanocellulose with widths ranging from 40 to 80 nm and lengths ranging from 0.1 to 1.8 μm. In this study, the CNCs with similar or improved dimensions could be produced by enzymatic hydrolysis with the EGs from *M. thermophila*.

In terms of CNC yield, it was found that the EG combinations of (5+7) and (5+7+45) produced a higher CNC yield than MtEG5A alone by enzymatic hydrolysis when using the same amount of total protein, indicating a synergistic effect between these enzymes during the enzymatic preparation of CNCs. The higher yield of CNCs obtained by the combination of (5+7+45) than the combination of (5+7) (8.80% vs. 6.40%) may be related to EG45 being a less processive endoglucanase than EG5 and EG7 [28,33]. The low processive EG45 was more likely to randomly hydrolyze different cellulose chains, which is also demonstrated by the larger size (length) of the CNC particles produced by MtEG45A (Table 3). The EG combination of (5+12) showed a low CNC yield (5.67%), although it was slightly higher than MtEG5A alone at the same total protein dosage, which should be related to the low CNC yield obtained with MtEG12A alone (Figure 2b). Compared to the results reported in the literature [17,21], using the MtEG combinations to hydrolyze EDP produced similar or higher CNC yields, indicating its potential in the enzymatic production of CNCs.

### 2.5. Characteristics of CNCs

#### 2.5.1. Morphological Observation

Figure 3 shows TEM images of the CNCs obtained by enzymatic hydrolysis of EDP using MtEG5A and different EG combinations including (5+7), (5+12), and (5+7+45) at 50 °C for 72 h. The presence of short rod-like structure with uniformity in size was found, which is characteristic of CNCs. Enzyme types and preparation methods affect CNC morphology; for example, Bondancia et al. [39] observed that the CNCs obtained from sugarcanes bagasse through hydrolysis with sulfuric acid (S-CNC), citric acid (Cit-CNC), and a combination of citric and sulfuric acids (Cit-S-CNC) displayed different morphological characteristics, with thin, elongated, acicular rod-like structures were observed for the S-CNC, and shorter nanocrystals for Cit-CNC. Additionally, association with acid also influences CNC morphology. Zhu et al. [40] produced spherical CNCs from microcrystalline cellulose by mixed acid hydrolysis. In another study, CNFs produced from bleached kraft softwood pulp using endoglucanases presented an intertwined and three-dimensional network structure morphology [24].

#### 2.5.2. XRD and Crystallinity Analysis

The high crystallinity of CNCs is an important property for biocomposite applications [13]. The XRD curves and crystallinity indexes (CrIs) of the CNCs obtained by enzymatic hydrolysis with MtEG5A and several EG combinations are shown in Figure 4. All the CNCs displayed similar peaks at 2θ = 16° (110), 22° (200), and 34° (004), as shown in Figure 4a, which can be attributed to the typical structure of cellulose I [41].

As shown in Figure 4b, the CrI values of the CNCs obtained by several EG combinations were slightly higher than that by MtEG5A alone, indicating that the EG combinations had a greater degradation effect on amorphous cellulose. The CNCs obtained by the EG combination of (5+7+45) showed a higher CrI value compared to other CNC samples, which may be attributed to EG45 being less processive [28,33].

#### 2.5.3. FTIR Analysis

The FTIR spectra of the CNCs obtained by enzymatic hydrolysis with MtEG5A and several EG combinations are shown in Appendix A. The samples peak observed at 3400 cm^−1^ were attributed to the bending and stretching vibrations of the OH- group [42,43]. The peaks observed at 2900 cm^−1^ correspond to C–H stretching vibrations [44,45]. The absorption band at 1646 cm^−1^ shows C-O-C asymmetric valence vibrations in cellulose and hemicellulose, and the peak at 890 cm^−1^ corresponds to C-O-C stretching of *β*-glycosidic linkages between glucose units in the cellulose. Also, the peak in the region of 1401 cm^−1^ and 1131 cm^−1^ present in the spectra were attributed to the C–H deformation (asymmetric) and O–H association bands in cellulose, respectively [46]. The FTIR spectra showed that the characteristic absorption peaks for all the CNCs were similar, with some specific differences due to the effect of the enzyme on the material components. The absorbance peaks for all samples indicated that the CNC particles in the suspension mainly consisted of a stable cellulose structure [47].

#### 2.5.4. TG Analysis

Thermal stability is an important property for the processing and application of CNCs. Hence, the obtained CNCs from enzymatic hydrolysis of EDP using MtEG5A and different combinations of MtEGs were analyzed by thermogravimetric analysis (Figure 5). According to TG analysis, the first region below 150 °C was attributed to water evaporation. The second region was associated with the decomposition of hemicellulose at temperatures between 200 and 315 °C, followed by the decomposition of cellulose from around 310 to 400 °C. On the other hand, small portions of lignin in the biomass decomposes over a broader degradation temperature range (250–700 °C) than the cellulose and hemicellulose components due to its aromatic ring structure [48]. In Figure 5a, it can be found that, for all the CNCs, there was only a slight weight loss between 100 and 150 °C due to the evaporation of absorbed water. As EDP contains almost no lignin, all the CNCs from EDP mainly presented a one-step degradation at a temperature between 250 °C and 400 °C caused by a small amount of hemicellulose and major cellulose degradation [49].

Table 4 shows the temperatures at which the biomasses started to decompose (T_onset_), the temperatures at which the mass degradation rates are maximal (T_max_), and the residual mass percentages at 600 °C. It was found that the CNCs obtained by enzymatic hydrolysis with MtEG5A alone had a relatively low T_onset_ and T_max_ compared to the CNCs obtained with the EG combinations. The T_onset_ values of CNCs obtained with the combinations of (5+7), (5+12), and (5+7+45) were found to be 262.31 °C, 257.25 °C, and 263.95 °C, respectively, which agree with previously reported studies [50,51]. An increased T_max_ for the CNC samples obtained with the EG combinations compared to MtEG5A alone was probably due to the more efficient removal of the disorder region in cellulose because of the synergistic effect between different EGs, which was consistent with the changes in CrI of the CNC samples, as shown in Figure 4a. It has been reported that the thermal stability of the sample corresponds to its crystallinity [52].

## 3. Materials and Methods

### 3.1. Materials and Strains

The EDP used is a commercial pulp. Pure sodium carboxymethylcellulose (Na-CMC) was purchased from Sigma-Aldrich to measure the activity of endoglucanases. All reagents used in this study were of analytical grade. *Myceliophthora thermophila* ATCC 42464 stock cultures were maintained on agar slants containing 1.5% malt extract agar, at 45 °C. Genomic DNA from *M. thermophila* was prepared and isolated according to the procedure described previously [53].

### 3.2. Construction of Engineering Strains for Heterologous Expression of EGs

The cDNA fragment of MtEG5A, MtEG7A, MtEG12A, and MtEG45A was amplified from genomic DNA by PCR reactions with gene-specific primers and Phanta Max Super-Fidelity DNA Polymerase (Vazyme Biotech, Nanjing, China). The amplified gene fragment contains a C-terminal 6× His tag sequence and signal peptide coding regions, which were predicted by SignalP (http://www.cbs.dtu.dk/services/SignalP/) (accessed on 10 March 2023), and was inserted into the BamHI site of P*ubiD-pyrG* to produce gene-specific expression vectors. The vector contains a P*ubiD* constitutive promoter from *P. oxalicum*, T*trpc*-Terminatior of *Aspergillus nadulus*, and the *pyrG*-OMP decarboxylase gene [31]. The cloning was performed using the ClonExpress II One Step Cloning Kit (Vazyme Biotech, Nanjing, China) according to the manufacturer’s instructions. DH5α *Escherichia coli* (Tsingke, Beijing, China) was used for plasmid transformation and amplification. The correct plasmid was verified by colony PCR with primers P*ubiD*-F and P*ubiD*-R, and the linear expression cassettes were then transformed into *P. oxalicum* M12 by the PEG-mediated protoplast transformation method as described previously [53]. The transformants were screened on a glucose agar media plate at 30 °C for 2–3 d. Then, the target genes were identified by PCR with primer pairs and sequencing. All the primers and their uses are listed in Appendix A.

### 3.3. Enzyme Production

The recombinant strain was inoculated into seed culture media (glucose 2%, peptone 0.1%, Vogal’s salt 2%) for 24 h at 30 °C, 200 rpm; then, 500 μL of the medium was transferred into fermentation medium (glucose 2%, Vogal’s salt 2%) and cultured at 30 °C, 200 rpm for 48 h. After culturing, the supernatant was obtained by centrifuged at 4 °C, 5000 rpm for 20 min and used as a crude enzyme sample. The concentration of protein was measured by the Bradford method with BSA (Sangon Biotech, Shanghai, China) as the standard [54]. For protein purification, the supernatant was filtered using 0.22 μm polyethersulfone membranes and passed through a 1 mL HisTrap™ FF column (GE Healthcare, Pittsburgh, PA, USA). The protein solution with a high concentration of imidazole was placed in a dialysis bag for 24 h. Dialysis was carried out at 4 °C, as described previously [55]. After that, the protein was carefully removed and transferred into a 5 mL EP tube and stored at 4 °C. Sodium dodecyl sulfate polyacrylamide gel electrophoresis (SDS-PAGE) of the crude enzyme and purified proteins was carried out using the method described by Yang et al. [17].

### 3.4. Effect of Temperature and pH on Enzymatic Activity

The enzymatic activity of the recombinant enzyme was determined using 1% Na-CMC at a wide range of pH values and temperatures to studying their effects on enzymatic activity. The optimum temperature of the purified protein was examined by measuring the enzymatic activity of the protein in the temperature range of 50–75 °C with 5 °C intervals. For thermostability analysis, the enzyme aliquots were exposed to 50, 55, 60, and 65 °C, for different time intervals, followed by a standard assay. To determine the effect of pH on enzyme activity, 1% Na-CMC substrate was first prepared using 0.1 M sodium citrate (pH 3.0–5.0), 0.1 M sodium phosphate (pH 6.0–7.0), or 0.1 M tris-HCl (pH 8.0–9.0) buffer. The purified enzyme was incubated in the substrate solutions prepared with different pHs and reacted for 30 min at the optimum temperature to determine the optimum pH.

### 3.5. CNC Preparation

CNCs were prepared by enzymatic hydrolysis using EDP as feedstock according to the procedure described previously [17], with some modifications. In brief, EDP with 4 mL sodium acetate buffer (50 mM, pH 4.8) was treated with ultrasound (KQ2200DE, Kunshan Ultrasonic Instruments Co., Ltd. Suzhou, China) at 100 W for 30 min. Then, the enzyme and buffer were added to the reaction system as required, and enzymatic hydrolysis was carried out in a 50 mL flask with a working volume of 15 mL under the conditions of enzyme dosages of 30 mg protein/g DM, solid content of 5% (w/w), 50 °C, 180 rpm for 72 h. After hydrolysis, the samples were boiled for 10 min to denature the enzyme and then filtered through a 0.22 μm membrane for separating the solid residue and liquid. The solid residue was added to 45 mL of distilled water and treated with ultrasound at 100 W for 30 min, and whirlpool-shaken to disperse the particles evenly, then centrifuged at 2600× *g* for 10 min (Eppendorf, 54197R), and the supernatants were collected into a 500 mL beaker as CNC solutions; this step was repeated five times to complete the collection of the CNCs.

### 3.6. Characterization of CNCs

#### 3.6.1. Dynamic Light Scattering (DLS)

The particle size distributions of the CNC samples were measured using a Zeta Plus analyzer (Brookhaven, New York, America) following the DLS principle. Measurements were carried out at 25 °C at a detection angle of 90° and performed in triplicate for error analysis.

#### 3.6.2. TEM Analysis

TEM was used to observe the microscopic morphology of the CNC samples. The CNC suspensions were placed on a copper mesh for sample preparation, and then dried at room temperature before analysis.

#### 3.6.3. SEM Observation

The CNC sample suspension of 0.07 wt% was dropped on aluminum foil, dried, and then coated with gold spray and observed at accelerating voltage and magnification of 5.00 kV and 50,000×, respectively, using SEM (Hitachi S-4800). ImageJ software was used to assess the length and width of the CNCs.

#### 3.6.4. XRD Analysis

XRD analysis of the CNCs was conducted using an X-ray diffractometer (Ultima IV, Rigaku, Tokyo, Japan) operated with CuKα radiation, generated at a voltage of 40 kV and current of 40 mA. Scans were carried out in a 2θ range from an angle of 5° to 50° at a scanning rate of 5°/min. The CrI was obtained by dividing the summed area of the crystalline peaks by the total area (including the areas under the crystalline and amorphous peaks [39,56].

#### 3.6.5. FT-IR Analysis

Infrared spectra were measured with diffuse reflectance (Nicolet iS10 model, Thermo Fisher Scientific Instrument, Madison, WI, USA) between 4000 and 400 cm−1, with a resolution of 4 cm−1 and scanned 16 times.

#### 3.6.6. Thermogravimetric Analysis (TGA)

The thermal stability of the CNC samples was analyzed using a Thermogravimetric analyzer (TG209F3 Tarsus, Germany). The dried samples (10–20 mg) were heated from 30 to 900 °C at a rate of 10 °C/min in a nitrogen atmosphere [57].

### 3.7. Analytical Method

#### 3.7.1. Chemical Compositions of EDP

Chemical components of the EDP, including the contents of cellulose, hemicellulose, and lignin, were determined according to the relevant methods published by the National Renewable Energy Laboratory (NREL) [58].

#### 3.7.2. Protein Concentration

The Bradford method was used to measure protein concentration. The standard curve was determined according to the procedure described previously [54]; in brief, 1 mg/mL BSA was diluted with distilled water to different concentrations in triplicate. Then, 20 μL of the diluted BSA was added to 200 μL of Bradford reagent (Sangon Biotech, Shanghai, China), and the absorbance (OD value) was measured using an ultraviolet spectrophotometer (UV-2550, Shimadzu Co., Kyoto, Japan) at a wavelength of 595 nm. Distilled water was used as the control. The standard curve was plotted as protein concentration (mg/mL) vs. ΔOD value.

#### 3.7.3. Enzyme Activity Assay

The standard enzymatic assay was performed in 50 mM sodium acetate buffer (pH 4.8) containing 1% Na-CMC and 20 µg/mL protein at 50 °C for 30 min; the reducing sugar produced was determined using the DNS method [59]. One unit (U) of endoglucanase activity was defined as the amount of enzyme that liberates 1 μmol glucose per min.

#### 3.7.4. CNC Yield

The yield of CNCs obtained was determined by drying the suspension of CNCs using the fridge dryer method. The yield of CNCs was calculated according to Equation (1).
(1)$$Y(\%)=\frac{M2}{M1}\times 100\%$$
where Y is the yield of CNCs (%), *M*1 is the initial weight of the EDP in the reaction system (g), and *M*2 is the final weight (dried) of the CNCs obtained (g). All experiments were performed in triplicate. The data is shown as the average in the figures, and error bars represent the standard deviation.

## 4. Conclusions

The four families of EGs from *M. thermophila* could serve as potential enzyme sources for the enzymatic preparation of CNCs by enzymatic hydrolysis of EDP, especially MtEG5A which has a high optimal reaction temperature (75 °C) and good thermal stability, and is expected to accelerate the enzymatic preparation of CNCs through enzymatic hydrolysis at high reaction temperatures. Compared to MtEG5A alone, enzymatic hydrolysis with combinations of MtEG5A and other EGs from *M. thermophila* such as the combination of 5+7+45 produced higher yields and improved CNC properties such as small particle size, high crystallinity, and thermal stability, indicating a synergistic effect between different GH families of endoglucanases during the enzymatic preparation of CNCs from EDP. This study provides a reference for understanding the enzymatic characteristics of different families of EGs from *M. thermophila* and their potential application in nanocellulose production.

## Figures and Tables

**Figure 1 ijms-24-10676-f001:**
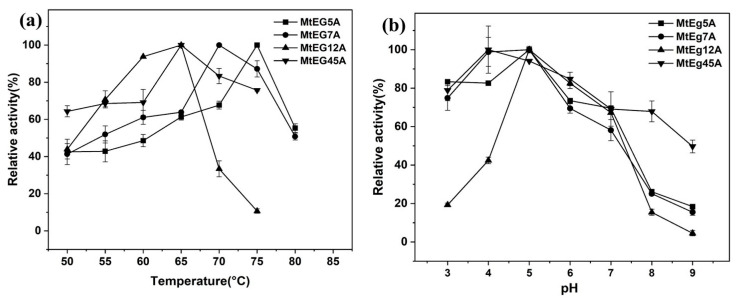
Effect of temperature (**a**) and pH (**b**) on the activity of different endoglucanases.

**Figure 2 ijms-24-10676-f002:**
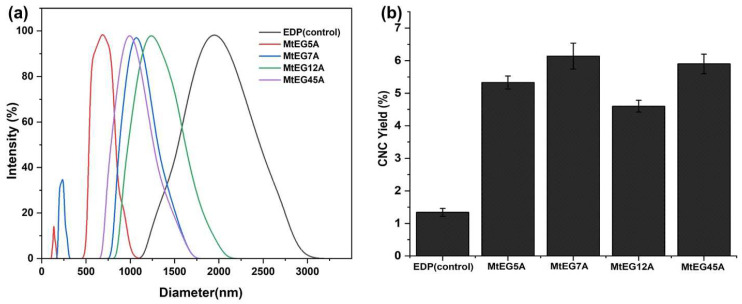
DLS analysis (**a**) and yield (**b**) of CNCs obtained by enzymatic hydrolysis of EDP with different EGs.

**Figure 3 ijms-24-10676-f003:**
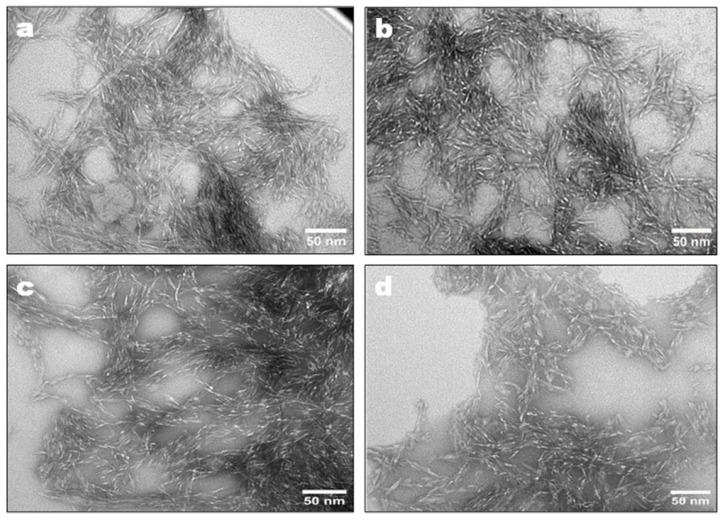
TEM images of CNCs prepared by enzymatic hydrolysis of EDP using MtEG5A (**a**), (5+7) (**b**), (5+12) (**c**), and (5+7+45) (**d**).

**Figure 4 ijms-24-10676-f004:**
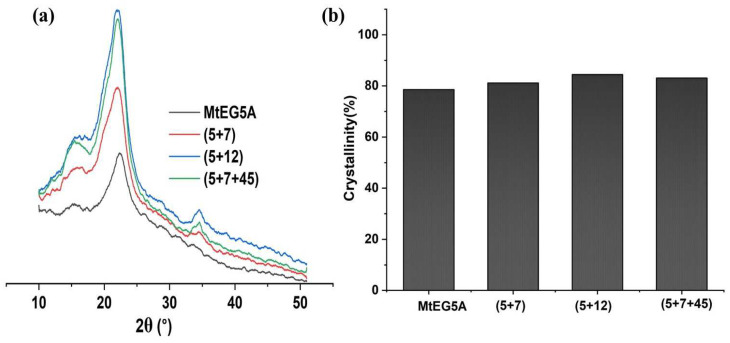
XRD pattern (**a**) and crystallinity (**b**) of CNCs prepared by enzymatic hydrolysis with MtEG5A alone and different EG combinations.

**Figure 5 ijms-24-10676-f005:**
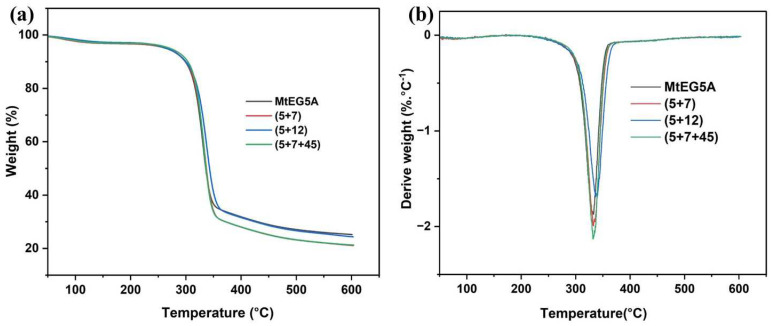
Thermal stability analysis of the CNCs: TGA curves (**a**); DTG curves (**b**).

**Table 1 ijms-24-10676-t001:** Chemical compositions of eucalyptus dissolving pulp * (%).

Cellulose	Hemicellulose	Lignin	Others
91.45 ± 0.12	2.23 ± 0.02	0.72 ± 0.03	5.08 ± 0.13

*: oven-dried weight of pulp.

**Table 2 ijms-24-10676-t002:** General information and specific activity of endoglucanases from *M. thermophila*.

Enzyme	GH Family	Accession No.	Predicated Molecular Weight (kDa)	CBM	Glycosylation		Specific Activity (U/mg)
					*N*	*O*	
MtEG5A	5	Mycth_86753	42.39	CBM	3	17	4.194
MtEG7A	7	Mycth_111372	48.67	CBM	2	16	3.60
MtEG12A	12	Mycth_109444	27.11	-	-	-	1.47
MtEG45A	45	Mycth_76901	23.64	-	-	3	2.51

**Table 3 ijms-24-10676-t003:** Characteristics and yields of CNCs obtained by enzymatic hydrolysis of EDP with MtEG5A and different combinations of MtEGs.

Enzyme Used in Preparation	Length * (nm)	Width * (nm)	Average Apparent Size (nm) **	CNCs Yield (%)
MtEG5A	825 ± 108	79 ± 16	696 ± 49	5.33 ± 0.2
(5+7)	386 ± 85	45 ± 13	476 ± 52	6.40 ± 0.16
(5+12)	466 ± 199	47 ± 7	633 ± 29	5.67 ± 1.1
(5+7+45)	609 ± 103	40 ± 5	670 ± 35	8.80 ± 1.2

Note: (1) * data were obtained by SEM; ** data were obtained by DLS analyses. (2) For all the experiments, the total protein dosage in the reaction system was 30 mg protein/g DM, and the proportion between different proteins was equal for each EG combination.

**Table 4 ijms-24-10676-t004:** T_onset_ and T_max_ of obtained CNCs and residue at 600 °C using different enzymes.

Sample	T_onset_ (°C)	T_max_ (°C)	Residue at 600 °C (%)
MtEG5A	246.58	330.14	27.74
(5+7)	262.31	333.09	25.51
(5+12)	257.25	340.32	27.07
(5+7+45)	263.95	338.18	24.76

## Data Availability

The data produced in this study have been included in this manuscript.

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
