# Peer review of "Production and Characterization of Cellulose Nanocrystals from Eucalyptus Dissolving Pulp Using Endoglucanases from Myceliophthora thermophila"

_ijms, 2023, doi:10.3390/ijms241310676_

Round 1
Reviewer 1 Report
A manuscript by Waghmare and colleagues presents the feasibility of novel enzymatic preparation of cellulose nanocrystals. The manuscript has some good basis, and their results represent an important contribution to the improvement of materials science in general. However, some of the issues listed below should be clarified before publication:
1) Throughout the manuscript, some words and phrases are written in italics with no clear meaning. Authors should thoroughly review the entire text and use italics only where necessary.
2) In the subsection 2.5.1 Morphological observation, the characterization of the different prepared CNCs is based only on TEM micrographs; however, the authors also mentioned that they were also characterised by SEM. The authors may also support their TEM results with SEM analysis.
3) In subsection 2.5.3. FTIR analysis, the authors should provide the corresponding FTIR spectra.
4) In the introduction, the authors also mention the shear-thinning behaviour in aqueous solutions. Therefore, I recommend that they perform additional rheological analyses that would contribute to a higher scientific value of the manuscript.
5) In subsection 3.6.3. SEM, the accelerating voltage and magnification at which the micrographs were taken?
6) The font and spacing between paragraphs should be consistent throughout the text.
Author Response
A manuscript by Waghmare and colleagues presents the feasibility of novel enzymatic preparation of cellulose nanocrystals. The manuscript has some good basis, and their results represent an important contribution to the improvement of materials science in general. However, some of the issues listed below should be clarified before publication:
1) Throughout the manuscript, some words and phrases are written in italics with no clear meaning. Authors should thoroughly review the entire text and use italics only where necessary.
Answer: Thank you for your comment. We have changed them according to your suggestion.
2) In the subsection 2.5.1 Morphological observation, the characterization of the different prepared CNCs is based only on TEM micrographs; however, the authors also mentioned that they were also characterized by SEM. The authors may also support their TEM results with SEM analysis.
Answer: Thank you for your comment. The results of SEM analysis of the CNCs have been shown and discussed in section 2.4 and Table 2. In supplementary files, SEM images of the CNCs were also provided in Fig.S3.
3) In subsection 2.5.3. FTIR analysis, the authors should provide the corresponding FTIR spectra.
Answer: Thank you for your comment. The FTIR spectra have been provided in Fig.S4 of supplementary files.
4) In the introduction, the authors also mention the shear-thinning behavior in aqueous solutions. Therefore, I recommend that they perform additional rheological analyses that would contribute to a higher scientific value of the manuscript.
Answer: Thank you for your comment. The present study was focused on evaluation of the effect of different endoglucanases and its combinations on CNC preparation to hope to screen suitable enzyme for high-efficiency production of CNC through enzymatic hydrolysis. Our results proved that it is feasible to use the endoglucanases from Myceliophthora thermophila to produce CNC. At present, however, the CNC yield is still relatively low, and the enzymatic hydrolysis efficiency still needs to be enhanced by different approaches such as process optimization, which is exactly what we are working on. After it, the characteristics of CNC with high yield, including rheological analyses, will be comprehensively evaluated for the further utilization of CNC.
5) In subsection 3.6.3. SEM, the accelerating voltage and magnification at which the micrographs were taken?
Answer: Thank you for your comment. The voltage and magnification were 5.00 kV and 50,000 x respectively, and it has been added in the Subsection “3.6.3 SEM observation”.
6) The font and spacing between paragraphs should be consistent throughout the text.
Answer: Thank you for your comment. We have checked this manuscript and revised issues with writing.
Reviewer 2 Report
Title:
Production and Characterization of Cellulose Nanocrystal from Eucalyptus Dissolving Pulp using Endoglucanases from My-celiophthora thermophila
Authors:
Pratima Waghmare, Nuo Xu, Pankajkumar Waghmare, Guodong Liu, Yinbo Qu, Xuezhi Li, Jian Zhao
Abstract:
Well composed.
Line 13 – Please change the word “showed” to “has”
Technical question:
1) Does your process is cost effective?? Is this feasible for commercialization??
2) Why did you not report the yield increase??
Introduction:
The overall content is well rounded and informative. However, it requires some minor editing. I really would like to see it revised.
Several names are reported in Italic format that are not necessary. i.e.,
Line – 29 (packaging) , 30 (antibacterial agents, and skincare, etc), 32, 33 (and bacterial nanocellulose (BNC), 42, 59, 73, 78, 88, etc….
Line 26 – 1-100nm or 1-100 nm – Spacing between the number and units. See line 38.
Line 42 – Spacing should be corrected. 300 °C and NOT 300° C.
Line 69 – Why the word (Beta) in line 56 (β-glucosidase) is different than the one in line 69 (ß-glucosidase)
Line 77 – Spacing - Should this be 30 % or 30%. See line 287 and 290. There is inconsistency.
Line 83 – I think the word CNCs should be written as CNC’s.
Lines 112-114 – Spacing – Please make sure the spacing between numbers and units are correct. i.e., 300°C or 300 °C?? also see line 349,
Lines 125- 127 – Spacing – Please insert one space between the names and the abbreviations like line 122. i.e., Transmission electron microscope (TEM) and not Transmission electron microscope(TEM).
Results and Discussion:
Several names are reported in Italic format that are not necessary. i.e.,
Line 135, 205, 207, 216,
Please respect spacing after the end of the sentence. i.e., in lines 127, 209, 319,
Line 143, The Asterisk (*) should appear on the word pulp and then with comma (,) then %.
Line 190 – see comment for spacing in the Introduction for Lines 112-114.
Line 193 – Same as Line 190
Line 221– is this Fig.2a or Fig. 2a??
Line 246 – is this Fig.S3 or Fig. S3??
Line 275 – this unit should be μm and NOT um.
Line 367 - please make sure the spacing between numbers and units are correct. There is inconsistency.
Line 369- Spacing between “increased Tmax for the CNC…
Line 375- Spacing – Please make sure the spacing between numbers and unit is correct.
Materials and Methods:
This section is very informative. Well done
Please make sure the spacing between numbers and units are correct. There are inconsistencies.
Line 446 - this unit should be μm and NOT um.
Line 447 – Spacing…
Lines 455 and 468 - there is inconsistency between numbers and units…
Equation (1) – Spacing
Conclusions:
It requires some minor editing.
Line 510- 75 should not be italic..
No mention of the extent of yield increase…Ha…
References:
Please make sure all the refences are consistent with the journal instructions.
The manuscript requires editing to improve consistency.
Author Response
Abstract:
Line 13 – Please change the word “showed” to “has”
Answer: Thank you for your comment. We have changed it.
Technical question:
1) Does your process is cost-effective?? Is this feasible for commercialization??
Answer: Thank you for your comment. Compared to common methods, the advantages of this enzymatic process include (1) no requirement for corrosion-resistant equipment, decreasing device cost; (2) being environmentally friendly, decreasing wastewater treatment cost, Especially for the treatment of highly difficult to treat concentrated acidic waste liquid; (3) milder operating conditions such as low reaction temperature (about 50 °C) with less energy cousmption compared to chemical and mechanical treatment,resulting in low operating cost; (4) the by-products that are dissolved sugars, which can be directly converted into liquid fuels and chemicals through enzymatic saccharification and fermentation, compensating for the process cost; and so on. Thus, Technically, this process is cost-effective, and is feasible for industrial production of CNCs. At present, the technical challenges of this process include the production of specialized commercial enzyme preparations used for CNC production, and the further improvement of CNC yield. By searching for more effective enzymes, as the work of this study, and optimizing enzyme combinations, as well as combined with process optimization, it is expected to solve these problems.
2) Why did you not report the yield increase??
Answer: Thank you for your comment. This study was focused on evaluation of the effect of different endoglucanases and its combinations on CNC preparation to hope to screen suitable enzyme (or enzyme mixture) for production of CNC through enzymatic hydrolysis. The work of this study proved the feasibility of the enzyme in application of CNCs production. Although the enzyme mixture such as (5+7) showed a better effect on preparation of CNC compared to single EG5, the increase in the CNC yield was relatively low. The process optimization for increasing the yield is conducted in our current work. Therefore, in this article, we did not emphasize the specific value of the increase in yield, but presented this phenomenon in the section of conclusion.
Introduction:
1) Several names are reported in Italic format that are not necessary. i.e.,
Line – 29 (packaging) , 30 (antibacterial agents, and skincare, etc), 32, 33 (and bacterial nanocellulose (BNC), 42, 59, 73, 78, 88, etc….
Answer: Thank you for your comment. Corrections have been made in the revised manuscript.
2) Line 26 – 1-100nm or 1-100 nm – Spacing between the number and units. See line 38.
Answer: Thank you for your comment. Corrections have been made in the revised manuscript.
3) Line 42 – Spacing should be corrected. 300 °C and NOT 300° C.
Answer: Thank you for your comment. Corrections have been made in the revised manuscript.
4) Line 69 – Why the word (Beta) in line 56 (β-glucosidase) is different than the one in line 69 (ß-glucosidase)
Answer: Thank you for your comment. We have modified the font format to make them unified.
5) Line 77 – Spacing - Should this be 30 % or 30%. See line 287 and 290. There is inconsistency.
Answer: Thank you for your comment. We have deleted this space and changed to 30%.
6) Line 83 – I think the word CNCs should be written as CNC’s.
Answer: Thank you for your comment. CNCs are the plural form of CNC (cellulose nanocrystal), abbreviated as cellulose nanocrystals. Generally, multiple nanocrystals are produced simultaneously during the production, rather than a single crystal.
7) Lines 112-114 – Spacing – Please make sure the spacing between numbers and units are correct. i.e., 300°C or 300 °C?? also see line 349,
Answer: Thank you for your letter, and we have deleted this spacing between numbers and units.
8) Lines 125- 127 – Spacing – Please insert one space between the names and the abbreviations like in line 122. i.e., Transmission electron microscope (TEM) and not Transmission electron microscope(TEM).
Answer: Thank you for your comment. Corrections have been made in the revised manuscript.
Results and Discussion:
Several names are reported in Italic format that are not necessary. i.e.,
Line 135, 205, 207, 216,
Answer: Thank you for your comment. Corrections have been made in the revised manuscript.
Please respect spacing after the end of the sentence. i.e., in lines 127, 209, 319,
Line 143, The Asterisk (*) should appear on the word pulp and then with comma (,) then %.
Answer: Thank you for your comment. Corrections have been made in the revised manuscript.
Line 190 – see comment for spacing in the Introduction for Lines 112-114 and Line 193.
Answer: Thank you for your comment. Corrections have been made in the revised manuscript.
Line 221– is this Fig.2a or Fig. 2a??
Answer: Thank you for your comment. This is Fig. 2a, and corrections have been made in the revised manuscript.
Line 246 – is this Fig.S3 or Fig. S3??
Answer: Thank you for your comment. This is Fig. S3, and corrections have been made in the revised manuscript.
Line 275 – this unit should be μm and NOT um.
Answer: Thank you for your comment. Corrections have been made in the manuscript.
Line 367 - please make sure the spacing between numbers and units are correct. There is inconsistency.
Answer: Thank you for your comment. Corrections have been made in the manuscript.
Line 369- Spacing between “increased Tmax for the CNC…
Answer: Thank you for your comment. Corrections have been made in the manuscript.
Line 375- Spacing – Please make sure the spacing between numbers and unit is correct.
Answer: Thank you for your comment. Corrections have been made in the manuscript.
Materials and Methods:
Please make sure the spacing between numbers and units are correct. There are inconsistencies.
Line 446 - this unit should be μm and NOT um.
Answer: Thank you for your comment. Corrections have been made in the manuscript.
22) Line 447 – Spacing…
Answer: Thank you for your comment. Corrections have been made in the manuscript.
Lines 455 and 468 - there is inconsistency between numbers and units…
Equation (1) – Spacing
Answer: Thank you for your comment. Corrections have been made in the manuscript.
Conclusions:
It requires some minor editing.
Line 510- 75 should not be italic..
Answer: Thank you for your comment. We have made minor modifications to the conclusion, including writing format.
No mention of the extent of yield increase…Ha…
Answer: Thank you for your comment. Here, we did not emphasize the specific extent of the increase in yield, and just presented this phenomenon of yield increase when using enzyme combinations instead of single enzyme EG5. The reason is that, although the enzyme mixture showed a better effect on preparation of CNC compared to single EG5, the increase in the CNC yield was still relatively low. The process optimization for increasing the yield is conducted in our current work.
References:
Please make sure all the references are consistent with the journal instructions.
Answer: Thank you for your comment. References are arranged as per journal’s instructions.
Comments on the Quality of English Language
The manuscript requires editing to improve consistency.
Answer: Thank you for your comment. Corrections have been made in the manuscript as per the reviewer’s suggestion.
Round 2
Reviewer 1 Report
The authors have successfully addressed my comments.